# SCALABLE NEURAL METHODS FOR
# REASONING WITH A SYMBOLIC KNOWLEDGE BASE

**William W. Cohen & Haitian Sun & R. Alex Hofer & Matthew Siegler**
Google, Inc
{wcohen,haitiansun,rofer,msiegler}@google.com

## ABSTRACT

We describe a novel way of representing a symbolic knowledge base (KB) called a *sparse-matrix reified KB*. This representation enables neural KB inference modules that are fully differentiable, faithful to the original semantics of the KB, expressive enough to model multi-hop inferences, and scalable enough to use with realistically large KBs. The sparse-matrix reified KB can be distributed across multiple GPUs, can scale to tens of millions of entities and facts, and is orders of magnitude faster than naive sparse-matrix implementations. The reified KB enables very simple end-to-end architectures to obtain competitive performance on several benchmarks representing two families of tasks: KB completion, and learning semantic parsers from denotations.

## 1 INTRODUCTION

There has been much prior work on using neural networks to *generalize* the contents of a KB (Xiong et al., 2017; Bordes et al., 2013; Dettmers et al., 2018), typically by constructing low-dimensional embeddings of the entities and relations in the KB, which are then used to score potential triples as plausible or implausible elements of the KB. We consider here the related but different problem of *incorporating a symbolic KB into a neural system*, so as to inject knowledge from an existing KB directly into a neural model. More precisely, we consider the problem of designing neural KB inference modules that are (1) fully differentiable, so that any loss based on their outputs can be backpropagated to their inputs; (2) accurate, in that they are faithful to the original semantics of the KB; (3) expressive, so they can perform non-trivial inferences; and (4) scalable, so that realistically large KBs can be incorporated into a neural model.

To motivate the goal of incorporating a symbolic KB into a neural network, consider the task of learning neural semantic parsers from denotations. Many questions—e.g., *what's the most recent movie that Quentin Tarantino directed?* or *which nearby restaurants have vegetarian entrees and take reservations?*—are best answered by *knowledge-based question-answering* (KBQA) methods, where an answer is found by accessing a KB. Within KBQA, a common approach is *neural semantic parsing*—i.e., using neural methods to translate a natural-language question into a structured query against the KB (Zhong et al., 2017; Finegan-Dollak et al., 2018; Shaw et al., 2019), which is subsequently executed with a symbolic KB query engine. While this approach can be effective, it requires training data pairing natural-language questions with structured queries, which is difficult to obtain. Hence researchers have also considered *learning semantic parsers from denotations* (Berant et al., 2013; Yih et al., 2015), where training data consists of pairs $(q, A)$, where $q$ is a natural-language question and $A$ is the desired answer. Typically $A$ is a set of KB entities—e.g., if $q$ is the first sample question above, $A$ would be[1] the singleton set containing *Once Upon a Time in Hollywood*.

Learning semantic parsers from denotations is difficult because the end-to-end process to be learned includes a non-differentiable operation—i.e., reasoning with the symbolic KB that contains the answers. To circumvent this difficulty, prior systems have used three different approaches. Some have used heuristic search to infer structured queries from denotations (Pasupat & Liang, 2016; Dasigi et al., 2019): this works in some cases but often an answer could be associated with many possible structured queries, introducing noise. Others have supplemented gradient approaches with

---

[1]At the time of this writing.

| $x$: an entity | $X$: weighted set of entities | $\mathbf{x}$: vector encoding $X$ | $N_E$: # entities in KB |
|---|---|---|---|
| $r$: an relation | $R$: weighted set of relations | $\mathbf{r}$: vector encoding $R$ | $N_R$: # relations in KB |
| $\mathbf{M}_r$: matrix for $r$ | $\mathbf{M}_R$: weighted sum of $\mathbf{M}_r$'s, see Eq 1 | $follow(\mathbf{x}, \mathbf{r})$: see Eq 2 | $N_T$: # triples in KB |
| $\mathbf{M}_{subj}, \mathbf{M}_{obj}, \mathbf{M}_{rel}$: the reified KB, encoded as matrices mapping triple id $\ell$ to subject, object, and relation ids | | | |

Table 1: Summary of notation used in the paper. (This excludes notation used in defining models for the KB completion and QA tasks of Section 3.)

reinforcement learning (e.g., (Misra et al., 2018)). Some systems have also "neuralized" KB reasoning, but to date only over *small* KBs: this approach is natural when answers are naturally constrained to depend on a small set of facts (e.g., a single table (Zhong et al., 2017; Gupta & Lewis, 2018)), but more generally requires coupling a learner with some (non-differentiable) mechanism to retrieve an appropriate small question-dependent subset of the KB as in (Sun et al., 2018; 2019).

In this paper, we introduce a novel scheme for incorporating reasoning on a large question-independent KB into a neural network, by representing a symbolic KB with an encoding called a *sparse-matrix reified KB*. A sparse-matrix reified KB is very compact, can be distributed across multiple GPUs if necessary, and is well-suited to modern GPU architecture. For KBs with many relations, a reified KB can be up to four orders of magnitude faster than alternative implementations (even alternatives based on sparse-matrix representations), and in our experiments we demonstrate scalability to a KB with over 13 million entities and nearly 44 million facts. This new architectural component leads to radically simpler architectures for neural semantic parsing from denotations—architectures based on a single end-to-end differentiable process, rather than cascades of retrieval and neural processes.

We show that very simple instantiations of these architectures are still highly competitive with the state of the art for several benchmark tasks. *To our knowledge these models are the first fully end-to-end neural parsers from denotations that have been applied to these benchmark tasks.* We also demonstrate that these architectures scale to long chains of reasoning on synthetic tasks, and demonstrate similarly simple architectures for a second task, KB completion.

## 2 NEURAL REASONING WITH A SYMBOLIC KB

### 2.1 BACKGROUND

**KBs, entities, and relations.** A KB consists of *entities* and *relations*. We use $x$ to denote an entity and $r$ to denote a relation. Each entity has an integer index between 1 and $N_E$, where $N_E$ is the number of entities in the KB, and we write $x_i$ for the entity that has index $i$. A relation is a set of entity pairs, and represents a relationship between entities: for instance, if $x_i$ represents "Quentin Tarantino" and $x_j$ represents "Pulp Fiction" then $(x_i, x_j)$ would be an member of the relation *director_of*. A relation $r$ can thus be represented as a subset of $\{1, \ldots, N_E\} \times \{1, \ldots, N_E\}$. Finally a KB consists a set of relations and a set of entities.

**Weighted sets as "$k$-hot" vectors.** Our differentiable operations are based on *weighted sets*, where each element $x$ of weighted set $X$ is associated with a non-negative real number. It is convenient to define this weight to be zero for all $x \notin X$, while for $x \in X$, a weight less than 1 is a confidence that the set contains $x$, and weights more than 1 make $X$ a multiset. If all elements of $X$ have weight 1, we say $X$ is a *hard set*. A weighted set $X$ can be encoded as an *entity-set vector* $\mathbf{x} \in \mathbb{R}^{N_E}$, where the $i$-th component of $\mathbf{x}$ is the weight of $x_i$ in $X$. If $X$ is a hard entity set, then this will be a "$k$-hot" vector, for $k = |X|$. The set of indices of $\mathbf{x}$ with non-zero values is called the *support of $x$*.

**Sets of relations, and relations as matrices** Often we would like to reason about sets of relations[2], so we also assume every relation $r$ in a KB is associated with an entity and hence an integer index. We write $r_k$ for the relation with index $k$, and we assume that relation entities are listed first in the index of entities, so the index $k$ for $r_k$ is between 1 and $N_R$, where $N_R$ is the number of relations in the KB. We use $R$ for a set of relations, e.g., $R = \{writer\_of, director\_of\}$ might be such a set, and use $\mathbf{r}$ for a vector encoding of a set. A relation $r$ can be encoded as a *relation matrix* $\mathbf{M}_r \in \mathbb{R}^{N_E \times N_E}$, where the value for $\mathbf{M}_r[i, j]$ is (in general) the weight of the assertion $r(x_i, x_j)$ in the KB. In the experiments of this paper, all KB relations are hard sets, so $\mathbf{M}_r[i, j] \in \{0, 1\}$.

---

[2]This is usually called *second-order* reasoning.

**Sparse vs. dense matrices for relations**. Scalably representing a large KB requires careful consideration of the implementation. One important issue is that for all but the smallest KBs, a relation matrix must be implemented using a *sparse matrix* data structure, as explicitly storing all $N_E^2$ values is impractical. For instance, consider a KB containing 10,000 movie entities and 100,000 person entities. A relationship like *writer_of* would have only a few tens of thousands of facts (since most movies have only one or two writers), but a dense matrix would have 1 billion values.

We thus model relations as *sparse matrices*. Let $N_r$ be the number of entity pairs in the relation $r$: common sparse matrix data structures require space $O(N_r)$. One common sparse matrix data structure is a *sparse coordinate pair (COO)* encoding: with a COO encoding, each KB fact requires storing only two integers and one float.

Our implementations are based on Tensorflow (Abadi et al., 2016), which offers limited support for sparse matrices. In particular, driven by the limitations of GPU architecture, Tensorflow only supports matrix multiplication between a sparse matrix COO and a dense matrix, but not between two sparse matrices, or between sparse higher-rank tensors and dense tensors.

**Entity types.** It is often possible to easily group entities into disjoint sets by some notion of "type": for example, in a movie domain, all entities might be either of the type "movie", "person", or "movie studio". It is straightforward to extend the formalism above to typed sets of entities, and doing this can lead to some useful optimizations. We use these optimizations below where appropriate: in particular, relation-set vectors **r** are of dimension $N_R$, not $N_E$, in the sections below. The full formal extension to typed entities and relations is given in Appendix A.

## 2.2 REASONING IN A KB

**The relation-set following operation.** Note that relations can also be viewed as labeled edges in a *knowledge graph*, the vertices of which are entities. Adopting this view, we define the *r-neighbors* of an entity $x_i$ to be the set of entities $x_j$ that are connected to $x_i$ by an edge labeled $r$, i.e., *r-neighbors(x)* $\equiv \{x_j : (x_i, x_j) \in r\}$. Extending this to relation sets, we define

$$\text{R-neighbors(X)} \equiv \{x_j : \exists r \in R, x_i \in X \text{ so that } (x_i, x_j) \in r\}$$

Computing the $R$-neighbors of an entity is a single-step reasoning operation: e.g., the answer to the question $q =$ "*what movies were produced or directed by Quentin Tarantino*" is precisely the set $R$-neighbors(X) for $R = \{producer\_of, writer\_of\}$ and $X = \{Quentin\_Tarantino\}$. "Multi-hop" reasoning operations require nested $R$-neighborhoods, e.g. if $R' = \{actor\_of\}$ then $R'$-neighbors($R$-neighbors($X$)) is the set of actors in movies produced or directed by Quentin Tarantino.

We would like to approximate the $R$-neighbors computation with differentiable operations that can be performed on the vectors encoding the sets $X$ and $R$. Let **x** encode a weighted set of entities $X$, and let **r** encode a weighted set of relations. We first define $\mathbf{M}_R$ to be a weighted mixture of the relation matrices for all relations in $R$ i.e.,

$$\mathbf{M}_R \equiv \left(\sum_{k=1}^{N_R} \mathbf{r}[k] \cdot \mathbf{M}_{r_k}\right) \tag{1}$$

We then define the *relation-set following operation for x and r* as:

$$follow(\mathbf{x}, \mathbf{r}) \equiv \mathbf{x}\mathbf{M}_R = \mathbf{x}\left(\sum_{k=1}^{N_R} \mathbf{r}[k] \cdot \mathbf{M}_{r_k}\right) \tag{2}$$

As we will show below, this differentiable numerical relation-set following operation can be used as a neural component to perform certain types of logical reasoning. In particular, Eq 2 corresponds closely to the logical $R$-neighborhood operation, as shown by the claim below.

**Claim 1** *The support of follow($\boldsymbol{x}, \boldsymbol{r}$) is exactly the set of R-neighbors(X).*

A proof and the implications of this are discussed in Appendix B.

## 2.3 SCALABLE RELATION-SET FOLLOWING WITH A REIFIED KB

**Baseline implementations.** Suppose the KB contains $N_R$ relations, $N_E$ entities, and $N_T$ triples. Typically $N_R < N_E < N_T \ll N_E^2$. As noted above, we implement each $\mathbf{M}_r$ as a sparse COO matrix,

| Strategy | Definition | Batch? | Space complexity | # Operations | | |
|---|---|---|---|---|---|---|
| | | | | sp-dense matmul | dense + or $\odot$ | sparse + |
| naive mixing | Eq 1-2 | no | $O(N_T + N_E + N_R)$ | 1 | 0 | $N_R$ |
| late mixing | Eq 3 | yes | $O(N_T + bN_E + bN_R)$ | $N_R$ | $N_R$ | 0 |
| reified KB | Eq 4 | yes | $O(bN_T + bN_E)$ | 3 | 1 | 0 |

Table 2: Complexity of implementations of relation-set following, where $N_T$ is the number of KB triples, $N_E$ the number of entities, $N_R$ the number of relations, and $b$ is batch size.

so collectively these matrices require space $O(N_T)$. Each triple appears in only one relation, so $\mathbf{M}_R$ in Eq 1 is also size $O(N_T)$. Since sparse-sparse matrix multiplication is not supported in Tensorflow we implement $\mathbf{x}\mathbf{M}_R$ using dense-sparse multiplication, so $\mathbf{x}$ must be a dense vector of size $O(N_E)$, as is the output of relation-set following. Thus the space complexity of $follow(\mathbf{x}, \mathbf{r})$ is $O(N_T + N_E + N_R)$, if implemented as suggested by Eq 2. We call this the *naive mixing* implementation, and its complexity is summarized in Table 2.

Because Tensorflow does not support general sparse tensor contractions, it is not always possible to extend sparse-matrix computations to minibatches. Thus we also consider a variant of naive mixing called *late mixing*, which mixes the *output* of many single-relation following steps, rather than mixing the KB itself:

$$follow(\mathbf{x}, \mathbf{r}) = \sum_{k=1}^{N_R} (\mathbf{r}[k] \cdot \mathbf{x}\mathbf{M}_{r_k}) \tag{3}$$

Unlike naive mixing, late mixing can be extended easily to a minibatches (see Appendix C). Let $b$ be the batch size and $\mathbf{X}$ be a minibatch of $b$ examples $[\mathbf{x}_1; \ldots; \mathbf{x}_b]$: then this approach leads to $N_R$ matrices $\mathbf{X}\mathbf{M}_k$, each of size $O(bN_E)$. However, they need not all be stored at once, so the space complexity becomes $O(bN_E + bN_R + N_T)$. An additional cost of late mixing is that we must now sum up $N_R$ dense matrices.

**A reified knowledge base.** While semantic parses for natural questions often use small sets of relations (often singleton ones), in learning there is substantial uncertainty about what the members of these small sets should be. Furthermore, realistic wide-coverage KBs have many relations—typically hundreds or thousands. This leads to a situation where, at least during early phases of learning, it is necessary to evaluate the result of mixing very large sets of relations. When many relations are mixed, late mixing becomes quite expensive (as experiments below show).

An alternative is to represent each KB assertion $r_k(x_i, x_j)$ as a tuple $(i, j, k)$ where $i, j, k$ are the indices of $x_i, x_j$, and $r_k$. There are $N_T$ such triples, so for $\ell = 1, \ldots, N_T$, let $(i_\ell, j_\ell, k_\ell)$ denote the $\ell$-th triple. We define these sparse matrices:

$$\mathbf{M}_{subj}[\ell, m] \equiv \begin{cases} 1 & \text{if } m = i_\ell \\ 0 & \text{else} \end{cases} \quad \mathbf{M}_{obj}[\ell, m] \equiv \begin{cases} 1 & \text{if } m = j_\ell \\ 0 & \text{else} \end{cases} \quad \mathbf{M}_{rel}[\ell, m] \equiv \begin{cases} 1 & \text{if } m = k_\ell \\ 0 & \text{else} \end{cases}$$

Conceptually, $\mathbf{M}_{subj}$ maps the index $\ell$ of the $\ell$-th triple to its subject entity; $\mathbf{M}_{obj}$ maps $\ell$ to the object entity; and $\mathbf{M}_{rel}$ maps $\ell$ to the relation. We can now implement the relation-set following as below, where $\odot$ is Hadamard product:

$$follow(\mathbf{x}, \mathbf{r}) = (\mathbf{x}\mathbf{M}_{subj}^T \odot \mathbf{r}\mathbf{M}_{rel}^T)\mathbf{M}_{obj} \tag{4}$$

Notice that $\mathbf{x}\mathbf{M}_{subj}^T$ are the triples with an entity in $\mathbf{x}$ as their subject, $\mathbf{r}\mathbf{M}_{rel}^T$ are the triples with a relation in $\mathbf{r}$, and the Hadamard product is the intersection of these. The final multiplication by $\mathbf{M}_{obj}$ finds the object entities of the triples in the intersection. These operations naturally extend to minibatches (see Appendix). The reified KB has size $O(N_T)$, the sets of triples that are intersected have size $O(bN_T)$, and the final result is size $O(bN_E)$, giving a final size of $O(bN_T + bN_E)$, with no dependence on $N_R$.

Table 2 summarizes the complexity of these three mathematically equivalent but computationally different implementations. The analysis suggests that the reified KB is preferable if there are many relations, which is the case for most realistic KBs[3].

---

[3]The larger benchmark datasets used in this paper have 200 and 616 relations respectively.

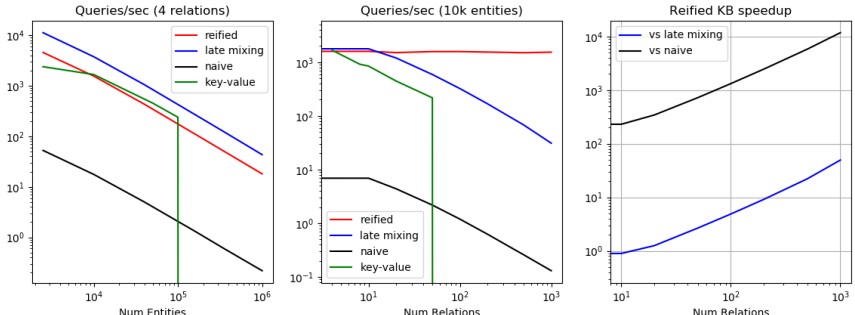

Figure 1: Left and middle: inference time in queries/sec on a synthetic KB as size and number of relations is varied. Queries/sec is given as zero when GPU memory of 12Gb is exceeded. Right: speedups of reified KBs over the baseline implementations.

**Distributing a large reified KB.** The reified KB representation is quite compact, using only six integers and three floats for each KB triple. However, since GPU memory is often limited, it is important to be able to *distribute* a KB across multiple GPUs. Although to our knowledge prior implementations of distributed matrix operations (e.g., (Shazeer et al., 2018)) do not support sparse matrices, sparse-dense matrix multiplication can be distributed across multiple machines. *We thus implemented a distributed sparse-matrix implementation of reified KBs*. We distibuted the matrices that define a reified KB "horizontally", so that different triple ids $\ell$ are stored on different GPUs. Details are provided in Appendix D.

## 3 EXPERIMENTS

### 3.1 SCALABILITY

Like prior work (Cohen et al., 2017; De Raedt et al., 2007), we used a synthetic KB based on an $n$-by-$n$ grid to study scalability of inference. Every grid cell is an entity, related to its immediate neighbors via relations *north*, *south*, *east*, and *west*. The KB for an $n$-by-$n$ grid thus has $O(n^2)$ entities and $O(n^2)$ triples. We measured the time to compute the 2-hop inference *follow*(*follow*($\mathbf{x}, \mathbf{r}$), $\mathbf{r}$) for minibatches of $b = 128$ one-hot vectors, and report it as queries per second (qps) on a single GPU (e.g., qps=1280 would mean a single minibatch requires 100ms). We also compare to a key-value memory network (Miller et al., 2016), using an embedding size of 64 for entities and relations, where there is one memory entry for every triple in the KB. Further details are given in Appendix E.

The results are shown Figure 1 (left and middle), on a log-log scale because some differences are very large. With only four relations (the leftmost plot), late mixing is about 3x faster than the reified KB method, and about 250x faster than the naive approach. However, for more than around 20 relations, the reified KB is faster (middle plot). As shown in the rightmost plot, the reified KB is 50x faster than late mixing with 1000 relations, and nearly 12,000x faster than the naive approach.

With this embedding size, the speed of the key-value network is similar to the reified KB for only four relations, however it is about 7x slower for 50 relations and 10k entities. Additionally, the space needed to store a triple is much larger in a key-value network than the reified KB, so memory is exhausted when the KB exceeds 200,000 entities (with four relations), or when the KB exceeds 100 relations (with 10,000 entities.) The reified KB scales much better, and can handle 10x as many entities and 20x as many relations.

### 3.2 MODELS USING REIFIED KBS

As discussed below in Section 4, the reified KB is closely related to key-value memory networks, so it can be viewed as a more efficient implementation of existing neural modules, optimized for reasoning with symbolic KBs. However, being able to include an entire KB into a model can lead to a *qualitative* difference in model complexity, since it is not necessary to build machinery to retrieve from the KB. To illustrate this, below we present simple models for several tasks, each using the reified KB in different ways, as appropriate to the task. We consider two families of tasks: learning semantic parsers from denotations over a large KB, and learning to complete a KB.

**KBQA for multi-hop questions.** MetaQA (Zhang et al., 2018) consists of 1.2M questions, evenly distributed into one-hop, two-hop, and three-hop questions. (E.g, the question "*who acted in a movie directed by Quentin Tarantino?*" is a two-hop question.) The accompanying KB (Miller et al., 2016) contains 43k entities and 186k triples. Past work treated one-hop, two-hop and three-hop questions separately, and the questions are labeled with the entity ids for the "seed entities" that begin the reasoning chains (e.g., the question above would be tagged with the id of the entity for *Quentin Tarantino*).

Using a reified KB for reasoning means the neural model only needs to predict the relations used at each stage in the reasoning process. For each step of inference we thus compute relation sets $\mathbf{r}^t$ using a differentiable function of the question, and then chain them together with relation-set following steps. Letting $\mathbf{x}^0$ be the set of entities associated with $q$, the model we use is:

$$\text{for } t = 1, 2, 3: \quad \mathbf{r}^t = f^t(q); \quad \mathbf{x}^t = follow(\mathbf{x}^{t-1}, \mathbf{r}^t)$$

where $follow(\mathbf{x}^{t-1}, \mathbf{r}^t)$ is implemented with a reified KB as described in Eq. 4.

To predict an answer on a $T$-hop subtask, we compute the softmax of the appropriate set $\mathbf{x}^T$. We used cross entropy loss of this set against the desired answer, represented as a uniform distribution over entities in the target set. Each function $f^t(q)$ is a different linear projection of a common encoding for $q$, specifically a mean-pooling of the tokens in $q$ encoded with a pre-trained 128-dimensional word2vec model (Mikolov et al., 2013). The full KB was loaded into a single GPU in our experiments.

It is interesting to contrast this simple model with the one proposed by Zhang et al. (2018). The "module for logic reasoning" they propose in Section 3.4 is fairly complex, with a description that requires a figure, three equations, and a page of text; furthermore, training this model requires constructing an example-dependent subgraph for each training instance. In our model, the "logic reasoning" (and all interaction with the KB) has been encapsulated completely in the $follow(\mathbf{x}, \mathbf{r})$ operation—which, as we will demonstrate below, can be re-used for many other problems. Encapsulating all KB reasoning with a single scalable differentiable neural module greatly simplifies modeling: in particular, *the problem of learning a structured KB query has been reduced to learning a few differentiable functions of the question*, one for each reasoning "hop". The learned functions are also interpretable: they are mixtures of relation identifiers which correspond to soft weighted sets of relations, which in turn softly specify which KB relation should be used in each stage of the reasoning process. Finally, optimization is simple, as the loss on predicted denotations can be back-propagated to the relation-prediction functions.

A similar modeling strategy is used in all the other models presented below.

**KBQA on FreeBase.** WebQuestionsSP (Yih et al., 2016) contains 4737 natural language questions, all of which are answerable using FreeBase (Bollacker et al., 2008), a large open-domain KB. Each question $q$ is again labeled with the entities $\mathbf{x}$ that appear in it.

FreeBase contains two kinds of nodes: real-world *entities*, and *compound value types* (CVTs), which represent non-binary relationships or events (e.g., a movie release event, which includes a movie id, a date, and a place.) Real-world entity nodes can be related to each other or to a CVT node, but CVT nodes are never directly related to each other. In this dataset, all questions can be answered with 1- or 2-hop chains, and all 2-hop reasoning chains pass through a CVT entity; however, unlike MetaQA, the number of hops is not known. Our model thus derives from $q$ three relation sets and then uniformly mixes both potential types of inferences:

$$\mathbf{r}_{\text{E} \to \text{E}} = f_{\text{E} \to \text{E}}(q); \quad \mathbf{r}_{\text{E} \to \text{CVT}} = f_{\text{E} \to \text{CVT}}(q); \quad \mathbf{r}_{\text{CVT} \to \text{E}} = f_{\text{CVT} \to \text{E}}(q)$$
$$\hat{\mathbf{a}} = follow(follow(\mathbf{x}, \mathbf{r}_{\text{E} \to \text{CVT}}), \mathbf{r}_{\text{CVT} \to \text{E}}) + follow(\mathbf{x}, \mathbf{r}_{\text{E} \to \text{E}})$$

We again apply a softmax to $\hat{\mathbf{a}}$ and use cross entropy loss, and $f_{\text{E} \to \text{E}}$, $f_{\text{E} \to \text{CVT}}$, and $f_{\text{CVT} \to \text{E}}$ are again linear projections of a word2vec encoding of $q$. We used a subset of Freebase with 43.7 million facts and 12.9 million entities, containing all facts in Freebase within 2-hops of entities mentioned in any question, excluding paths through some very common entities. We split the KB across three 12-Gb GPUs, and used a fourth GPU for the rest of the model.

This dataset is a good illustration of the scalability issues associated with prior approaches to including a KB in a model, such as key-value memory networks. A key-value network can be trained to implement something similar to relation-set following, if it stores all the KB triples in memory. If

we assume 64-float embeddings for the 12.9M entities, *the full KB of 43.7M facts would be 67Gb in size*, which is impractical. Additionally performing a softmax over the 43.7M keys would be prohibitively expensive, as shown by the experiments of Figure 1. This is the reason why in standard practice with key-value memory networks for KBs, the memory is populated with a heuristically subset of the KB, rather than the full KB. We compare experimentally to this approach in Table 3.

**Knowledge base completion**. Following Yang et al. (2017) we treat KB completion as an inference task, analogous to KBQA: a query $q$ is a relation name and a head entity $\mathbf{x}$, and from this we predict a set of tail entities. We assume the answers are computed with the disjunction of multiple inference chains of varying length. Each inference chain has a maximum length of $T$ and we build $N$ distinct inference chains in total, using this model (where $\mathbf{x}_i^0 = \mathbf{x}$ for every chain $i$):

$$\text{for } i = 1, \ldots, N \text{ and } t = 1, \ldots, T: \quad \mathbf{r}_i^t = f_i^t(q); \quad \mathbf{x}_i^t = follow(\mathbf{x}_i^{t-1}, \mathbf{r}_i^t) + \mathbf{x}_i^{t-1}$$

The final output is a softmax of the mix of all the $\mathbf{x}_i^T$'s: i.e., we let $\hat{\mathbf{a}} = softmax(\sum_{i \in \{1 \ldots N\}} \mathbf{x}_i^T)$. The update $\mathbf{x}_i^{t+1} = follow(\mathbf{x}_i^t, \mathbf{r}_i^t) + \mathbf{x}_i^t$ gives the model access to outputs of all chains of length less than $t$ (for more intuition see Appendix E.) The encoding of $q$ is based on a lookup table, and each $f_i^t$ is a learned linear transformation of $q$'s embedding.[4]

**An encoder-decoder architecture for varying inferential structures.** To explore performance on more complex reasoning tasks, we generated simple artificial natural-language sentences describing longer chains of relationships on a 10-by-10 grid. For this task we used an encoder-decoder model which emits chains of relation-set following operations. The question is encoded with the final hidden state of an LSTM, written here $\mathbf{h}^0$. We then generate a reasoning chain of length up to $T$ using a decoder LSTM. At iteration $t$, the decoder emits a scalar probability of "stopping", $p^t$, and a distribution over relations to follow $\mathbf{r}^t$, and then, as we did for the KBQA tasks, sets $\mathbf{x}^t = follow(\mathbf{x}^{t-1}, \mathbf{r}^t)$. Finally the decoder updates its hidden state to $\mathbf{h}^t$ using an LSTM cell that "reads" the "input" $\mathbf{r}^{t-1}$. For each step $t$, the model thus contains the steps

$$p^t = f_p(\mathbf{h}^{t-1}); \quad \mathbf{r}^t = f_r(\mathbf{h}^{t-1}); \quad \mathbf{x}^t = follow(\mathbf{x}^{t-1}, \mathbf{r}^t); \quad \mathbf{h}^t = \text{LSTM}(\mathbf{h}^{t-1}, \mathbf{r}^{t-1})$$

The final predicted location is a mixture of all the $\mathbf{x}_t$'s weighted by the probability of stopping $p_t$ at iteration $t$, i.e., $\hat{\mathbf{a}} = softmax(\sum_{t=1}^T \mathbf{x}^t \cdot p^t \prod_{t' < t}(1 - p^{t'}))$. The function $f_r$ is a softmax over a linear projection, and $f_p$ is a logistic function. In the experiments, we trained on 360,000 sentences requiring between 1 and $T$ hops and tested on an additional 12,000 sentences.

**Experimental results.** We next consider the performance of these models relative to strong baselines for each task. We emphasize our goal here is *not to challenge the current state of the art on any particular benchmark*, and clearly there are many ways the models of this paper could be improved. (For instance, our question encodings are based on word2vec, rather than contextual encodings (Devlin et al., 2018), and likewise relations are predicted with simple linear classifiers, rather than, say, attention queries over some semantically meaningful space, such as might be produced with language models or KB embedding approaches (Bordes et al., 2013)). Rather, our contribution is to present a generally useful scheme for including symbolic KB reasoning into a model, and we have thus focused on describing simple, easily understood models that do this for several tasks. However, it is important to confirm experimentally that the reified KB models "work"—e.g., that they are amenable to use of standard optimizers, etc.

Performance (using Hits@1) of our models on the KBQA tasks is shown in Table 3. For the non-synthetic tasks we also compare to a Key-Value Memory Network (KV-Mem) baseline (Miller et al., 2016). For the smaller MetaQA dataset, KV-Mem is initialized with all facts within 3 hops of the query entities, and for WebQuestionsSP it is initialized by a random-walk process seeded by the query entities (see (Sun et al., 2018; Zhang et al., 2018) for details). ReifKB consistently outperforms the baseline, dramatically so for longer reasoning chains. The synthetic grid task shows that there is very little degradation as chain length increases, with Hits@1 for 10 hops still 89.7%. It also illustrates the ability to predict entities in a KB, as well as relations.

We also compare these results to two much more complex architectures that perform end-to-end question answering in the same setting used here: VRN (Zhang et al., 2018), GRAFT-Net (Sun et al., 2018), and PullNet (Sun et al., 2019). All three systems build question-dependent subgraphs of the

---

[4]In the experiments we tune the hyperparameters $T \in \{1, \ldots, 6\}$ and $N \in \{1, 2, 3\}$ on a dev set.

|  | ReifKB (ours) | ReifKB + mask | KV-Mem (baseline) | VRN | GRAFT-Net | PullNet | non-differentiable components of architectures | |
|---|---|---|---|---|---|---|---|---|
| WebQSP | 52.7 | — | 46.7 | — | **67.8** | **68.1** | KV-Mem | initial memory retrieval |
| MetaQA | | | | | | | | |
| 1-hop | 96.2 | — | 95.8 | **97.5** | 97.0 | 97.0 | | |
| 2-hop | 81.1 | 95.4 | 25.1 | 89.9 | 94.8 | **99.9** | VRN | question-specific |
| 3-hop | 72.3 | 79.7 | 10.1 | 62.5 | 77.2 | **91.4** | GRAFTNet | subgraph retrieval |
| Grid | | | | | | | PullNet | all iterative retrievals |
| 5-hop | 98.4 | — | — | — | — | – | | |
| 10-hop | 89.7 | — | — | — | — | – | ReifKB(ours) | *none* |

Table 3: Hits@1 on the KBQA datasets. Results for KV-Mem and VRN on MetaQA are from (Zhang et al., 2018); results for GRAFT-Net, PullNet and KV-Mem on WebQSP are from (Sun et al., 2018) and (Sun et al., 2019).

KB, and then use graph CNN-like methods (Kipf & Welling, 2016) to "reason" with these graphs. Although not superior, ReifKB model is competitive with these approaches, especially on the most difficult 3-hop setting.

A small extension to this model is to mask the seed entities out of the answers (see Appendix E). This model (given as ReifKB + mask) has better performance than GRAFT-Net on 2-hop and 3-hop questions.

For KB completion, we evaluated the model on the NELL-995 dataset (Xiong et al., 2017) which is paired with a KB with 154k facts, 75k entities, and 200 relations. On the left of Table 4 we compare our model with three popular embedding approaches (results are from Das et al. (2017)). The reified KB model outperforms DistMult (Yang et al., 2014), is slightly worse than ConvE (Dettmers et al., 2018), and is comparable to ComplEx (Trouillon et al., 2017).

The competitive performance of the ReifKB model is perhaps surprising, since it has many fewer parameters than the baseline models—only one float and two integers per KB triple, plus a small number of parameters to define the $f_i^t$ functions for each relation. The ability to use fewer parameters is directly related to the fact that our model *directly uses inference on the existing symbolic KB* in its model, rather than having to learn embeddings that approximate this inference. Or course, since the KB is incomplete, some learning is still required, but learning is quite different: the system learns logical inference chains in the incomplete KB that approximate a target relation. In this setting for KBC, the ability to perform logical inference "out of the box" appears to be very advantageous.

Another relative disadvantage of KB embedding methods is that KB embeddings are generally *transductive*—they only make predictions for entities seen in training. As a non-transductive baseline, we also compared to the MINERVA model, which uses reinforcement learning (RL) methods to learn how to traverse a KB to find a desired answer. Although RL methods are less suitable as "neural modules", MINERVA is arguably a plausible competitor to end-to-end learning with a reified KB.

MINERVA slightly outperforms our simple KB completion model on the NELL-995 task. However, unlike our model, MINERVA is trained to find a *single answer*, rather than trained to infer a *set of answers*. To explore this difference, we compared to MINERVA on the grid task under two conditions: (1) the KB relations are the grid directions north, south, east and west, so the output of the target chain is always a *single* grid location, and (2) the KB relations also include a "vertical move" (north or south) and a "horizontal move" (east or west), so the result of the target chain can be a *set* of locations. As expected MINERVA's performance drops dramatically in the second case, from 99.3%

|  | NELL-995 | | |  | ReifKB (Ours) | MINERVA |
|---|---|---|---|---|---|---|
|  | H@1 | H@10 | | NELL-995 | 64.1 | **66.3** |
| ReifKB (Ours) | 64.1 | 82.4 | | Grid with seed entity | | |
| DistMult* | 61.0 | 79.5 | | 10-hop NSEW | 98.9 | **99.3** |
| ComplEx* | 61.2 | 82.7 | | 10-hop NSEW-VH | **73.6** | 34.4 |
| ConvE* | **67.2** | **86.4** | | MetaQA 3-hop | **72.3** | 41.7 |

Table 4: Left: Hits@1 and Hits@10 for KB completion on NELL 995. Starred KB completion methods are transductive, and do not generalize to entities not seen in training. Right: Comparison to MINERVA on several tasks for Hits@1.

|  | NELL-995 | MetaQA-3hop | WebQuestionsSP |
|---|---|---|---|
| # Facts | 154,213 | 196,453 | 43,724,175 |
| # Entities | 75,492 | 43,230 | 12,942,798 |
| # Relations | 200 | 9 | 616 |
| Time (seconds) | 44.3 | 72.6 | 1820 |

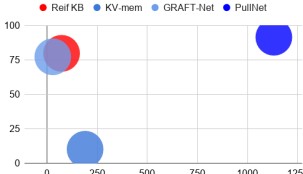

Table 5: Left, time to run 10K examples for KBs of different size. Right, time for 10k examples vs Hits@1 performance for ReifKB compared to three baselines on MetaQA-3hop questions.

Hits@1 to 34.4 %, while our model's performance is more robust. MetaQA answers can also be sets, so we also modified MetaQA so that MINERVA could be used (by making the non-entity part of the sentence the "relation" input and the seed entity the "start node" input) and noted a similarly poor performance for MINERVA. These results are shown on the right of Table 4.

In Tables 5 we compare the training time of our model with minibatch size of 10 on NELL-995, MetaQA, and WebQuestionsSP. With over 40 million facts and nearly 13 million entities from Freebase, it takes less than 10 minutes to run one epoch over WebQuestionsSP (with 3097 training examples) on four P100 GPUs. In the accompanying plot, we also summarize the tradeoffs between accuracy and training time for our model and three baselines on the MetaQA 3-hop task. (Here ideal performance is toward the upper left of the plot). The state-of-the-art PullNet Sun et al. (2019) system, which uses a learned method to incrementally retrieve from the KB, is about 15 times slower than the reified KB system. GRAFT-Net is only slightly less accurate, but also only slightly faster: recall that GRAFT-Net uses a heuristically selected subset (of up to 500 triples) from the KB for each query, while our system uses the full KB. Here the full KB is about 400 times as large as the question-specific subset used by GRAFT-Net. A key-value memory baseline including the full KB is nearly three times as slow as our system, while also performing quite poorly.

# 4 RELATED WORK

The relation-set following operation using reified KBs is implemented in an open-source package called NQL, for neural query language. NQL implements a broader range of operations for manipulating KBs, which are described in a companion paper (Cohen et al., 2019). This paper focuses on implementation and evaluation of the relation-set following operation with different KB representations, issues not covered in the companion paper.

TensorLog (Cohen et al., 2017), a probabilistic logic which also can be compiled to Tensorflow, and hence is another differentiable approach to neuralizing a KB. TensorLog is also based on sparse matrices, but does not support relation sets, making it unnatural to express the models shown in this paper, and does not use the more efficient reified KB representation. The differentiable theorem prover (DTP) is another differentiable logic (Rocktäschel & Riedel, 2017), but DPT appears to be much less scalable: it has not been applied to KBs larger than a few thousand triples. The Neural ILP system (Yang et al., 2017) uses approaches related to late mixing together with an LSTM controller to perform KB completion and some simple QA tasks, but it is a monolithic architecture focused on rule-learning, while in contrast we propose a re-usable neural component, which can be used in as a component in many different architectures, and a scalable implementation of this. It has also been reported that neural ILP does not scale to the size of the NELL995 task (Das et al., 2017).

The goals of this paper are related to KB embedding methods, but distinct. In KB embedding, models are generally fully differentiable, but it is not considered necessary (or even desirable) to accurately match the behavior of inference in the original KB. Being able to construct a learned *approximation* of a symbolic KB is undeniably useful in some contexts, but embedded KBs also have many disadvantages. In particular, they are much larger than a reified KB, with many more learned parameters—typically a long dense vector for every KB entity. Embedded models are typically evaluated by their ability to score a single triple accurately, and many models are not capable of executing multi-step KB inferences efficiently; further, models that do allow multi-step inference are known to produce cascaded errors on long reasoning chains (Guu et al., 2015; Hamilton et al., 2018). In contrast we focus on accurate models of reasoning in a symbolic KB, which requires consideration of novel scalability issues associated with sparse matrice representations.

Mathematically, our definition of relation-set following is much like the bilinear model for path following from Guu et al. (2015); however, we generalize this to path queries that include weighted sets of relations, allowing the relations in paths to be learned. Similar differences apply to the work of Hamilton et al. (2018), which extends the work of Guu et al. (2015) to include intersection operations. The vector representation used here for weighted sets in a reified KB makes intersection trivial to implement, as intersection corresponds to Hadamard product. Conveniently set union also corresponds to vector sum, and the complement of $X$ is $1 - \mathbf{x}$, which is perhaps why only a single additional neural operation is needed to support the KB reasoning tasks needed for the five benchmark tasks considered here.

Neural architectures like memory networks (Weston et al., 2014), or other architectures that use attention over some data structure approximating assertions (Andreas et al., 2016; Gupta & Lewis, 2018) can be used to build soft versions of relation-set following: however, they also do not scale well to large KBs, so they are typically used either with a non-differentiable *ad hoc* retrieval mechanism, or else in cases where a small amount of information is relevant to a question (Weston et al., 2015; Zhong et al., 2017). Similarly graph CNNs (Kipf & Welling, 2016) also can be used for reasoning, and often do use sparse matrix multiplication, but again existing implementations have not been scaled to tens of millions of triples/edges or millions of entities/graph nodes. Additionally, while graph CNNs have been used for reasoning tasks, the formal connection between them and logical reasoning remains unclear, whereas there is a precise connection between relation-set following and inference.

Reinforcement learning (RL) methods have been used to learn mappings from natural-language questions to non-differentiable logical representations (Liang et al., 2016; 2018) and have also been applied to KB completion tasks (Das et al., 2017; Xiong et al., 2017). Above we compared experimentally to MINERVA, one such method; however, the gradient-based approaches enabled by our methods are generally preferred as being easier to implement and tune on new problems, and easier to combine in a modular way with other architectural elements.

## 5 Conclusions

We introduced here a novel way of representing a symbolic knowledge base (KB) called a sparse-matrix reified KB. This representation enables neural modules that are fully differentiable, faithful to the original semantics of the KB, expressive enough to model multi-hop inferences, and scalable enough to use with realistically large KBs. In a reified KB, all KB relations are represented with three sparse matrices, which can be distributed across multiple GPUs, and symbolic reasoning on realistic KBs with many relations is much faster than with naive implementations—more than four orders of magnitude faster on synthetic-data experiments compared to naive sparse-matrix implementations.

This new architectural component leads to radically simpler architectures for neural semantic parsing from denotations and KB completion—in particular, they make it possible to learn neural KBQA models in a completely end-to-end way, mapping from text to KB entity sets, for KBs with tens of millions of triples and entities and hundreds of relations.

### Acknowledgments

The authors are greatful to comments and suggestions from Fernando Peireira, Bhuwan Dhingra, and many other colleagues on earlier versions of this work.

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

## A    ADDITIONAL BACKGROUND AND EXTENSIONS

**KBs, entities, and relations, and types.** In the more general case, a KB consists of *entities*, *relations*, and *types*. Again use $x$ to denote an entity and $r$ to denote a relation. We also assume each entity $x$ has a *type*, written $type(x)$, and let $N_\tau$ denote the number of entities of type $\tau$. Each entity $x$ in type $\tau$ has a unique index $index_\tau(x)$, which is an integer between 1 and $N_\tau$. We write $x_{\tau,i}$ for the entity that has index $i$ in type $\tau$, or $x_i$ if the type is clear from context.

Every relation $r$ has a *subject type* $\tau_{subj}$ and an *object type* $\tau_{obj}$, which constrain the types of $x$ and $x'$ for any pair $(x, x') \in r$. Hence $r$ can be encoded as a subset of $\{1, \ldots, N_{\tau_{subj}}\} \times \{1, \ldots, N_{\tau_{obj}}\}$. Relations with the same subject and object types are called *type-compatible*.

Our differentiable operations are based on *typed weighted sets*, where again each element $x$ of weighted set $X$ is associated with a non-negative real number, written $\omega[\![x \in X]\!]$, and we define $\omega[\![x \in X]\!] \equiv 0$ for all $x \notin X$. A set $X$ has a type $type(X) = \tau$, and all members of $X$ must be entities of type $\tau$.

We also assume every relation $r$ in a KB is associated with an entity $x_r$, and hence, an index and a type. Sets of relations $R$ are allowed only if all members are type-compatible. For example $R = \{writer\_of, director\_of\}$ might be a set of type-compatible relations.

A weighted set $X$ of type $\tau$ can be encoded as an *entity-set vector* $\mathbf{x} \in \mathbb{R}^{N_\tau}$, where the $i$-th component of $\mathbf{x}$ is the weight of the $i$-th entity of that type in the set $X$: e.g., $\mathbf{x}[index_\tau(x)] = \omega[\![x \in X]\!]$. We also use $type(\mathbf{x})$ to denote the type $\tau$ of the set encoded by $\mathbf{x}$.

A relation $r$ with subject type $\tau_1$ and object type $\tau_2$ can be encoded as a *relation matrix* $\mathbf{M}_r \in \mathbb{R}^{N_{\tau_1} \times N_{\tau_2}}$.

**Background on sparse matrices.** A COO encoding consists of a $N_r \times 2$ matrix $\mathbf{Ind}_r$ containing pairs of entity indices, and a parallel vector $\mathbf{w}_r \in \mathbb{R}^{N_r}$ containing the weights of the corresponding entity pairs. In this encoding, if $(i, j)$ is row $k$ of $\mathbf{Ind}$, then $\mathbf{M}_r[i, j] = \mathbf{w}_r[k]$, and if $(i, j)$ does not appear in $\mathbf{Ind}_r$, then $\mathbf{M}[i, j]$ is zero.

**Extension to soft KBs**. In the paper, we assume the non-zero weights in a relation matrix $\mathbf{M}_r$ are all equal to 1.0. This can be relaxed: if assertions in a KB are associated with confidences, then this confidence can be stored in $\mathbf{M}_r$. In this case, the reified KB must be extended to encode the weight for a triple: we find it convenient to redefine $\mathbf{M}_{rel}$ to hold that weight. In particular if the weight for the the $\ell$-th triple $r_k(x_i, x_j)$ is $w_\ell$, then we let

$$\mathbf{M}_{rel}[\ell, m] \equiv \begin{cases} w_\ell & \text{if } m = k_\ell \\ 0 & \text{else} \end{cases}$$

## B    PROOF OF CLAIM 1

**Claim 1** *The support of follow($\boldsymbol{x}, \boldsymbol{r}$) is exactly the set of R-neighbors(X).*

To better understand this claim, let $\mathbf{z} = follow(\mathbf{x}, \mathbf{r})$. The claim states $\mathbf{z}$ can approximate the $R$ neighborhood of any hard sets $R, X$ by setting to zero the appropriate components of $\mathbf{x}$ and $\mathbf{r}$. It is also clear that $\mathbf{z}[j]$ decreases when one decreases the weights in $\mathbf{r}$ of the relations that link $x_j$ to entities in $X$, and likewise, $\mathbf{z}[j]$ decreases if one decreases the weights of the entities in $X$ that are linked to $x_j$ via relations in $R$, so there is a smooth, differentiable path to reach this approximation.

More formally, consider first a matrix $\mathbf{M}_r$ encoding a single binary relation $r$, and consider the vector $\mathbf{x}' = \mathbf{x}\mathbf{M}_r$. As weighted sets, $X$ and $r$ have non-negative entries, so clearly for all $i$,

$$\mathbf{x}'[j] \neq 0 \;\text{ iff }\; \exists j : \mathbf{M}_r[i, j] \neq 0 \wedge \mathbf{x}[i] \neq 0 \;\text{ iff }\; \exists x_i \in X \text{ so that } (x_i, x_j) \in r$$

and so if $\mathbf{r}$ is a one-hot vector for the set $\{r\}$, then the support of *follow*($\mathbf{x}, \mathbf{r}$) is exactly the set $r$-neighbors($X$). Finally note that the mixture $\mathbf{M}_R$ has the property that $\mathbf{M}_R[i(e_1), i(e_2)] > 0$ exactly when $e_1$ is related to $e_2$ by some relation $r \in R$.

## C  MINIBATCHED COMPUTATIONS OF NAIVE AND LATE MIXING

The major problem with naive mixing is that, in the absence of general sparse tensor contractions, it is difficult to adapt to mini-batches—i.e., a setting in which $\mathbf{x}$ and $\mathbf{r}$ are replaced with matrices $\mathbf{X}$ and $\mathbf{R}$ with minibatch size $b$. An alternative strategy is *late mixing*, which mixes the *output* of many single-relation following steps, rather than mixing the KB itself:

$$follow(\mathbf{X}, \mathbf{R}) = \sum_{k=1}^{N_R} (\mathbf{R}[:, k] \cdot \mathbf{XM}_k)$$

Here $\mathbf{R}[:, k]$, the $k$-th column of $\mathbf{R}$, is "broadcast" to element of the matrix $\mathbf{XM}_k$. As noted in the body of the text, while there are $N_R$ matrices $\mathbf{XM}_k$, each of size $O(bN_E)$, they need not all be stored at once, so the space complexity becomes $O(bN_E + bN_R + N_T)$; however we must now sum up $N_R$ dense matrices.

The implementation of relation-set following for the reified KB can be straightforwardely extended to a minibatch:

$$follow(\mathbf{X}, \mathbf{R}) = (\mathbf{XM}_{subj}^T \odot \mathbf{RM}_{rel}^T)\mathbf{M}_{obj}$$

## D  DISTRIBUTED MATRIX MULTIPLICATION

Matrix multiplication $\mathbf{xM}$ was distributed as follows: $\mathbf{x}$ can be split into a "horizontal stacking" of $m$ submatrices, which we write as $[\mathbf{x}_1; \ldots; \mathbf{x}_m]$, and $\mathbf{M}$ can be similarly partitioned into $m^2$ submatrices. We then have the result that

$$\mathbf{xM} = [\mathbf{x}_1; \mathbf{x}_2; \ldots; \mathbf{x}_m] \begin{bmatrix} \mathbf{M}_{1,1} & \mathbf{M}_{1,2} & \ldots & \mathbf{M}_{1,m} \\ \vdots & \vdots & & \vdots \\ \mathbf{M}_{m,1} & \mathbf{M}_{m,2} & \ldots & \mathbf{M}_{m,m} \end{bmatrix} = \left[ (\sum_{i=1}^{m} \mathbf{x}_1 \mathbf{M}_{i,1}); \ldots; (\sum_{i=1}^{m} \mathbf{x}_m \mathbf{M}_{i,m}) \right]$$

This can be computed without storing either $\mathbf{X}$ or $\mathbf{M}$ on a single machine, and mathematically applies to both dense and sparse matrices. In our experiments we distibuted the matrices that define a reified KB "horizontally", so that different triple ids $\ell$ are stored on different GPUs.

Specifically, we shard the "triple index" dimension $N_T$ of matrices $\mathbf{M}_{subj}, \mathbf{M}_{rel}$ and $\mathbf{M}_{obj}$ in Eq. 4 to perform a distributed relation-set following on the reified KB. Let $\mathbf{M}_{subj,i}$ be the $i$'th shard of matrix $\mathbf{M}_{subj}$, and thus $\mathbf{M}_{subj} = [\mathbf{M}_{subj,1}^T; \ldots; \mathbf{M}_{subj,m}^T]^T \in \mathbb{R}^{N_T \times N_E}$. $\mathbf{M}_{obj}$ and $\mathbf{M}_{rel}$ are represented in the similar way. A distributed relation-set following is computed as a combination of relation-set following results on all shards of the KB.

$$follow(\mathbf{x}, \mathbf{r}) = (\mathbf{xM}_{subj}^T \odot \mathbf{rM}_{rel}^T)\mathbf{M}_{obj}$$

$$= \left( [\mathbf{xM}_{subj,1}^T; \ldots; \mathbf{xM}_{subj,m}^T] \odot [\mathbf{rM}_{rel,1}^T; \ldots; \mathbf{rM}_{rel,m}^T] \right) \begin{bmatrix} \mathbf{M}_{obj,1} \\ \vdots \\ \mathbf{M}_{obj,m} \end{bmatrix} \quad (5)$$

$$= \sum_{i=1}^{m} (\mathbf{xM}_{subj,i}^T \odot \mathbf{rM}_{rel,i}^T)\mathbf{M}_{obj,i} \quad (6)$$

This method can be easily extended to a mini-batch of examples $\mathbf{X}$.

## E  EXPERIMENTAL DETAILS

**Reproducing experiments.**  To reproduce these experiments, first download and install the Google `language` package[5]. Many of the experiments in this paper can be reproduced using scripts stored in the some subdirectory of the source directory `language/nql/demos`: for example, the scalability experiments of Figure 1 can be performed using scripts in `language/nql/demos/gridworld_scaling/`.

---

[5] `https://github.com/google-research/language.git`

**Grid experiments.** In the grid experiments, the entity vector $\mathbf{x}$ is a randomly-chosen singleton set, and the relation vector $\mathbf{r}$ weights relations roughly uniformly—more specifically, each relation has weight $1+\epsilon$ where $\epsilon$ is a drawn uniformly at random between 0 and 0.001.[6] We vary the number of relations by inventing $m$ new relation names and assigning existing grid edges to each new relation. These experiments were conducted on a Titan Xp GPU with 12Gb of memory.

For key-value networks, the key is the concatenation of a relation and a subject entity, and the value is the object entity. We considered only the run-time for queries on an untrained randomly-initialized network (since run-time performance on a trained network would be the same); however, it should be noted that considerable time that might be needed to train the key-value memory to approximate the KB. (In fact, it is not obvious under what conditions a KB can be approximated well by the key-value memory.)

We do not show results on the grid task for smaller minibatch sizes, but both reified and late mixing are about 40x slower with $b = 1$ than with $b = 128$.

**WebQuestionsSP experiments.** For efficiency, on this problem we exploit the type structure of the problem (see Appendix A). Our model uses two types of nodes, CVT and entity nodes. The model also uses three types of relations: relations mapping entities to entities, relations mapping entities to CVT nodes; and relations mapping CVT nodes to entity nodes.

**MetaQA experiments.** An example of a 2-hop question in MetaQA could be "Who co-starred with Robert Downey Jr. in their movies?", and the answer would be a set of actor entities, e.g., "Chris Hemsworth", "Thomas Stanley", etc. Triples in the knowledge base are represented as (subject, relation, object) triples, e.g., ("Robert Downey Jr.", "act_in", "Avengers: Endgame"), ("Avengers: Endgame", "stars", "Thomas Stanley"), etc. The quoted strings here all indicate KB entities.

We also observed that in the MetaQA 2-hop and 3-hop questions, the questions often exclude the seed entities (e.g., "other movies with the same director as Pulp Fiction"). This can be modeled by masking out seed entities from the predictions after the second hop (ReifKB + mask in the table).

**Timing on MetaQA and other natural problems.** The raw data for the bubble plot of Table 5 is below.

| Time (seconds) | Accuracy (hits@1) | Method |
|---|---|---|
| 72.6 | 79.7 | Reif KB |
| 189.8 | 10.1 | KV-mem |
| 28.9 | 77.2 | GRAFT-Net |
| 1131.0 | 91.4 | PullNet |

**Discussion of the KB completion model.** The KB completion model is

$$\text{for } i = 1, \ldots, N \text{ and } t = 1, \ldots, T: \quad \mathbf{r}_i^t = f_i^t(q); \quad \mathbf{x}_i^t = follow(\mathbf{x}_i^{t-1}, \mathbf{r}_i^t) + \mathbf{x}_i^{t-1}$$

It may not be immediately obvious why we used

$$\mathbf{x}_i^t = follow(\mathbf{x}_i^{t-1}, \mathbf{r}_i^t) + \mathbf{x}_i^{t-1}$$

instead of the simpler

$$\mathbf{x}_i^t = follow(\mathbf{x}_i^{t-1}, \mathbf{r}_i^t)$$

In the main text, we say that this "gives the model access to outputs of all chains of length less than $t$". This statement is probably easiest to understand by considering a concete example. Let us simplify notation slightly by dropping the subscripts and writing $follow(\mathbf{x}_i^{t-1}, \mathbf{r}_i^t)$ as $f^t(\mathbf{x}^{t-1})$. Now expand the definition of $\mathbf{x}^t$ for a few small values of $t$, using the linearity of the definition of relation-set

---

[6]If the relation weights do not vary from trial to trial, some versions of Tensorflow will optimize computation by precomputing and caching the matrix $\mathbf{M}_R$ from Eq. 1, which speeds up the naive method considerably. Of course, this optimization is impossible when learning relation sets.

following where appropriate to simplify:

$$\mathbf{x}^1 = f^1(\mathbf{x}^0) + \mathbf{x}^0$$
$$\mathbf{x}^2 = f^2(\mathbf{x}^1) + \mathbf{x}^1$$
$$= f^2\big((f^1(\mathbf{x}^0) + \mathbf{x}^0\big) + \big((f^1(\mathbf{x}^0) + \mathbf{x}^0\big)$$
$$= f^2(f^1(\mathbf{x}^0)) + f^2(\mathbf{x}^0) + f^1(\mathbf{x}^0) + \mathbf{x}^0$$
$$\mathbf{x}^3 = f^3(\mathbf{x}^2) + \mathbf{x}^2$$
$$= f^3\big((f^2(f^1(\mathbf{x}^0)) + f^2(\mathbf{x}^0) + f^1(\mathbf{x}^0) + \mathbf{x}^0\big) + f^2(f^1(\mathbf{x}^0)) + f^2(\mathbf{x}^0) + f^1(\mathbf{x}^0) + \mathbf{x}^0$$
$$= f^3(f^2(f^1(\mathbf{x}^0))) + f^3(f^2(\mathbf{x}^0)) + f^3(f^1(\mathbf{x}^0)) + f^3(\mathbf{x}^0) + f^2(f^1(\mathbf{x}^0)) + f^2(\mathbf{x}^0) + f^1(\mathbf{x}^0) + \mathbf{x}^0$$

A pattern is now clear: with this recursive definition $\mathbf{x}^t$ expands to a mixture of many paths, each of which applies a different subset of $f^1, \ldots, f^t$ to the initial input $\mathbf{x}$. Since the weights of the mixture can to a large extent be controlled by varying the norm of the relation vectors $\mathbf{r}^1, \ldots, \mathbf{r}^t$, this "kernel-like trick" increases the expressive power of the model without introducing new parameters. The final mixture of the $\mathbf{x}^t$'s seems to provide a bias towards accepting the output of shorter paths, which appears to be useful in practice.

