# OpenReview forum: "Scalable Neural Methods for Reasoning With a Symbolic Knowledge   Base"
_ICLR.cc/2020/Conference — Accept (Poster)_

### Official Review · AnonReviewer1 · 2019-10-21
**Official Blind Review #1**

**Rating:** 6

**Review:**

The paper provides a way to represent symbolic KBs called sparse matrix reified. Relations and entities' types are modelled using sparse matrices. This modelization allows distributing the computation on many GPUs. A neural model is used to manage these matrices. The trained model can be used to perform multi-hop queries. The proposed system has been compared with related systems. The results show that it achieves hits@1 that are comparable with the others.

The proposed approach seems promising, however, I feel that the paper is not ready for publication. The experiments lack a scalability test with related systems, the scalability test included in the paper only takes into account other definitions of the system. Also, comparison on the run time should be performed to see if the lower performance in terms of hits@1 (which are good anyway) is balanced by a better run time.

Therefore, it is difficult to see whether the proposed system is good or not.

As for the description of the system, it seems to me to be quite foggy. In my opinion, the neural model should be better described to show how the sparse matrices are mapped into the model.
Also, how are the training and testing sentences created? How would the output of a query look?

After reading the paper I have the feeling that it was written in a bit of a hurry, without working on the details. There are several typos, parentheses not correctly opened or closed, out of place commas that make some sentences seem unrelated to the rest of the paragraph. However, here last problems are easily fixable.

To sum up, I am conflicted about the choice of the final score. On the one hand, the approach is interesting, on the other hand the article seems to me not mature enough and strong enough from the point of view of organization, contents and experimental results shown.

Minor problems
On page 4, the size of the matrices XM_k uses the factor b that is introduced later.
In the introduction, next to footnote 1, I suggest specifying that A is the first query, the one about Tarantino's movies.
Also, his surname is misspelt throughout the paper. The correct one is Tarantino.

**Experience Assessment:**

I have read many papers in this area.

**Review Assessment: Checking Correctness Of Derivations And Theory:**

N/A

**Review Assessment: Checking Correctness Of Experiments:**

N/A

**Review Assessment: Thoroughness In Paper Reading:**

N/A

---

> ### Author Response · Authors · 2019-11-14
> **Response to Blind Review #1**
>
> Thank you for your thoughtful review.  We respond below to the your
> comments.
>
> RESPONSE TO "experiments lack a scalability test": We made two major
> changes in the revised paper.
>
> 1) We compared to key-value memory networks in Figure 1, demonstrating
> that reified KBs are much faster for many relations, and much more
> memory-efficient.
>
> 2) We compared the run-time efficiency of ReifKB with several baseline
> models on the metaQA tasks. The performance and running time on 10k
> samples are shown in the table below, which is also given in the
> Appendix, and a bubble chart figure is presented in Table 5.
>
> Time (sec)  (hits@1)  	Method
> ---------------------------------
> 72.6	 	 79.7		 Reif KB
> 189.8	    10.1		 KV-mem
> 28.9	    	 77.2		 GRAFT-Net
> 1131	    91.4      	PullNet
>
>
> In summary compared to Key-Value Memory Network (KV-mem), ReifKB is
> more efficient in run-time and achieves much better
> performance. GRAFT-Net is a pipeline approach which runs some
> heuristic-based retrieval algorithm to retrieve 500 triples from the
> KB for each question in MetaQA separately and then applies an
> expensive Graph-CNN model to find answers. It’s encouraging to see
> that ReifKB is only two times slower than GRAFT-Net while running on a
> KB which is 400 times larger!  PullNet is the state-of-the-art system
> on MetaQA recently proposed in EMNLP 2019 that includes a complicated
> learned iterative process of retrieval and classification. PullNet has
> better performance than ReifKB but is significantly slower.
>
> RESPONSE TO "the neural model should be better described to show how
> the sparse matrices are mapped into the model."  Briefly, the
> connection to the sparse matrices is given by the follow(x,r)
> operation that appears in each model.  To clarify this in the
> revision, we
>
> 1) Discuss the application to neural modeling briefly when we define
> the follow operation in Eq 2;
>
> 2) Mention the connection to the follow operation after the first
> model we present (for MetaQA)
>
> 3) Include a paragraph of additional explanation after the first model
> we present.
>
> RESPONSE TO "Also, how are the training and testing sentences created?
> How would the output of a query look?" We added some examples to the
> appendix.  For example, a 2-hop question in MetaQA could be “Who
> co-start with Robert Downey Jr. in their movies?”, and the answers is
> a list of actors (entities) “Chris Hemsworth”, “Thomas Stanley”,
> etc. Triples in the knowledge base are represented as <subject,
> relation, object>, such as, <Robert Downey Jr., act_in, Avengers:
> Endgame>, <Avengers: Endgame, stars, Thomas Stanley>, <Avengers:
> Endgame, published_in, 2019>, etc.
>
> RESPONSE TO: "There are several typos, ...": We have fixed the minor
> problems you noticed --- thank you! --- and believe the revised
> version is much clearer.
>
> RESPONSE TO: "To sum up, I am conflicted about the choice of the final
> score. On the one hand, the approach is interesting, on the other hand
> the article seems to me not mature enough and strong enough from the
> point of view of organization, contents and experimental results
> shown."  Thanks for finding the work interesting - we hope the many
> changes in the revision will improve your view of the paper.

---

### Official Review · AnonReviewer2 · 2019-10-22
**Official Blind Review #2**

**Rating:** 6

**Review:**

The paper proposes sparse-matrix KB representation for end-to-end KB reasoning tasks. They demonstrate that their algorithm is scalable to large knowledge graphs which is the central contribution of the paper.
They apply this to a bunch of tasks such as KB Completion and KBQA. This is done by mapping the query to a set (weights) of relations over which reasoning is performed.

I would first like to comment that I found the paper very hard to read. Thus due to my difficulty in understanding the paper, it is possible that I might have misunderstood parts of the paper.

The notations are overly complex. In my opinion, the notations can be simplified to a considerable extent. I would suggest a Table of notations or a small figure explaining the model. The paper, in my view, requires  considerable rewriting.

The paper states that the proposed approach encodes three floating point values and 6 integers for each triple. Is this true for KBC task? Because I am quite surprised that ReifKB approaches SOTA which use hundreds of floats for representing each entity and relation (e.g. DistMult/ComplEx).

Here are somethings which are not very clear to me KBQA tasks: It is not fully clear how the query is mapped to r. In my understanding r is a set of relations. Is the output of linear function taken as weights?
KBC Task: How do you ensure that the N chains are distinct? Do you have N different linear functions (f_i)? Also why is x_i^t added to follow in KBC task?

The paper needs considerable rewriting and therefore I cannot recommend this paper for acceptance at this stage.

**Experience Assessment:**

I have published one or two papers in this area.

**Review Assessment: Checking Correctness Of Derivations And Theory:**

I assessed the sensibility of the derivations and theory.

**Review Assessment: Checking Correctness Of Experiments:**

I assessed the sensibility of the experiments.

**Review Assessment: Thoroughness In Paper Reading:**

I read the paper at least twice and used my best judgement in assessing the paper.

---

> ### Author Response · Authors · 2019-11-14
> **Response to Blind Review #2**
>
> Thank you for your thoughtful review.  We respond below to the your
> comments and try and answer your questions.
>
> RESPONSE TO "notations are overly complex": We thank you for the
> comment and believe the paper is now much clearer.  In the revised
> version, we made a number of simplifying changes.  We removed all
> discussion of entity and relation types, and removed the omega[x in X]
> notation and the delta[a=b] notation. We put the discussion of types
> which are used mainly in the WebQuestionsSP experiment, into an
> appendix, and also moved a discussion of how to use soft KBs into an
> appendix.  We made the definition of M_R into a separate Equation.  We
> also relegated some details, such as minibatching and how distributed
> matrix multiplication is performed, to appendices.  As requested, we
> added a table of notation.  ONLY 15 NOTATIONAL CONVENTIONS ARE NOW
> DEFINED IN THE BODY OF THE PAPER including the convention of calling
> entities x.  (The several task-specific models we introduce still
> require some notation to define, but the use of this notation is very
> local and shouldn't cause much confusion to the reader.)
>
> RESPONSE TO "I am quite surprised that ReifKB approaches SOTA which
> use hundreds of floats for representing each entity and relation":
> Yes, this is also true for the KBC task!  It’s a little surprising but
> one of the strengths of the method.  The reason for the difference is
> that ReifKB doesn’t try and produce a generalizable embedded version
> of every entity in the KB - instead it builds a soft version of a
> symbolic KB query that approximates the target relation, so the model
> that it constructs is only a few parameters for each relation it is
> learning about.  We added the following discussion in the revision,
> when we discuss the KBC results, to emphasize this: "The competitive
> performance of the ReifKB model is perhaps surprising, since it has
> many fewer parameters than the baseline models---only one float and
> two integers per KB triple, plus a small number of parameters to
> define the $f_i^t$ functions for each relation.  The ability to use
> fewer parameters is directly related to the fact that our model
> \emph{directly uses inference on the existing symbolic KB} in its
> model, rather than having to learn embeddings that approximate this
> inference.  Or course, since the KB is incomplete, some learning is
> still required, but for KBC the ability to perform inference on the
> incomplete KB ``out of the box'' appears to be very advantageous."
>
> RESPONSE TO SPECIFIC QUESTIONS:
>
> Q: Here are somethings which are not very clear to me KBQA tasks: It is
> not fully clear how the query is mapped to r. In my understanding r is
> a set of relations. Is the output of linear function taken as weights?
>
> A: Your interpretation is exactly right: r is a vector of weights over
> relations, which are then used as vectors \textbf{r} in Eq (1) ---
> i.e., they are interpreted as weights used to mix the sparse-matrix
> encoding of the relations in the KB.
>
> Q: KBC Task: How do you ensure that the N chains are distinct? Do you
> have N different linear functions (f_i)?
>
> A: The N chains don’t need to be distinct: if you don’t need all N
> chains to model a relation the optimizer could chose to duplicate
> them.  We do indeed have multiple linear functions - a total of N*T of
> them.
>
> Q: Also why is x_i^t added to follow in KBC task?
>
> A: Why adding x_i^t works is difficult to answer succinctly: briefly,
> expanding the sum out recursively means that the model includes many
> diffent types of paths through the KB.  We added a final subsection of
> Appendix E that gives some intuition of why we this is true.

---

### Official Review · AnonReviewer3 · 2019-10-22
**Official Blind Review #3**

**Rating:** 6

**Review:**

This paper proposes an efficient sparse-matrix based representation for symbolic knowledge bases. This representation enables fully differentiable neural modules to model multi-hop inferences, which is designed to be scalable to handle realistically large knowledge bases. Experiments using the proposed method with end-to-end architectures on downstream KB tasks demonstrate its effectiveness and efficiency. Overall, this paper makes clear contributions and can inspire other researchers in the community to apply this KB representation for various learning / reasoning tasks on knowledge bases. However, considering the readability, I would like to recommend a weak accept for this paper.

Advantages of this paper: 1) The proposed method employs three sparse matrices to represent all KB relations, which is more efficient than existing work such as TensorLog and can support relation sets; 2) The reified KB representation is scalable, which can be distributed across multiple GPUs, making it much faster than the naive implementation; 3) The proposed method can be naturally used in end-to-end neural models and efficiently trained with gradient-based approaches.

Disadvantages of this paper: 1) In the introduction part, the authors mention that existing neural KB reasoning methods generally require some non-differentiable mechanism to retrieve small question-dependent subset of the KB, but there exist some existing methods such as memory networks that are fully differentiable for the KBQA task. It would be better to cover and discuss more existing methods when summarizing their properties; 2) In the methodology part (Section 2), the description is clear but lacks some guidance for the readers to understand the motivation behind the scene. For example, it would be helpful to discuss why representing relations as sparse matrices is necessary, whether there is any other choices, and the benefit of making the current choice. This might seem obvious for the authors, but can help readers that are not familiar with the context access the methodology more easily and understand the motivation better.

**Experience Assessment:**

I have published one or two papers in this area.

**Review Assessment: Checking Correctness Of Derivations And Theory:**

I carefully checked the derivations and theory.

**Review Assessment: Checking Correctness Of Experiments:**

I carefully checked the experiments.

**Review Assessment: Thoroughness In Paper Reading:**

I read the paper thoroughly.

---

> ### Author Response · Authors · 2019-11-14
> **Response to Blind Review #3**
>
> Thank you for your thoughtful review.  We respond below to the
> disadvantages you list.
>
> RESPONSE TO DISADVANTAGE 1) "... cover and discuss more existing
> methods:" In the first submission, we did discuss this briefly in the
> related work section, saying: "Neural architectures like memory
> networks (Weston et al., 2014), or other architectures that use
> attention over some data structure approximating assertions (Andreas
> et al., 2016; Gupta & Lewis, 2018) can be used to build soft versions
> of relation-set following: however, they also do not scale well to
> large KBs, so they are typically used either with a non-differentiable
> ad hoc retrieval mechanism, or else in cases where a small amount of
> information is relevant to a question (e.g., (Weston et al., 2015;
> Zhong et al., 2017))."
>
> In the revision we extended the discussion in related works of
> embedding methods and how our work is different.  Also, we added more
> material to the experimental section to further clarify the relation
> to key-value networks.  We plotted time in qps for a key-value network
> in Figure 1, and verified that runtime memory consumption is much
> worse than the reified KB---the model we used can only be used on 1/20
> the relations and 1/10 the triples before completely exhausting a 12Gb
> GPU memory.  We added this text to Section 3.2, where we discuss
> webQuestionsSP: This dataset [webQuestionsSP] is a good illustration
> of the scalability issues associated with prior approaches to
> including a KB in a model, such as key-value memory networks.  A
> key-value network that appends relations and subject entities as a key
> and has an object embedding as the value could be trained to implement
> something similar to relation-set following.  If we assume 64-float
> embeddings for the 12.9M entities, the full KB of 43.7M facts would be
> 67Gb in size, which is impractical, and a softmax over the 43.7M keys
> would be prohibitively expensive.  We provide more detailed timing
> information for key-value networks on MetaQA in a new figure in Table
> 5 (the precise numbers are given in Appendix E).
>
> RESPONSE TO DISADVANTAGE 2) "[the description] lacks some guidance
> for the readers to understand the motivation behind the scene. For
> example, it would be helpful to discuss why representing relations as
> sparse matrices is necessary".  The original paper discussed this
> briefly, and partly in a footnote: we did say in sec 2.1: "For all but
> the smallest KBs, a relation matrix must be implemented using a sparse
> matrix data structure, as explicitly storing all Nτ1 × Nτ2 values is
> impractical [4]" with footnote 4 saying: "For instance, if a KB
> contains 10,000 movie entities and 100,000 person entities, then a
> relationship like writer_of would require storing 1 billion values—far
> more than few tens of thousands of writer_of facts that would be in
> the KB (since most movies have only one or writers.)"
>
> In the revision, we expanded this discussion to a paragraph as
> follows: "Scalably representing a large KB requires careful
> consideration of the implementation.  One important issue is that for
> all but the smallest KBs, a relation matrix must be implemented using
> a sparse matrix data structure, as explicitly storing all N_E x N_E
> values is impractical.  For instance, consider a KB containing 10,000
> movie entities and 100,000 person entities.  A relationship like
> writer_of would have only a few tens of thousands of facts, since most
> movies have only one or two writers, but a dense matrix would have
> more than 1 billion values."
>
> We also give more concrete experimental comparisons to key-value
> memory networks, which are a dense-matrix alternative to the reified
> KB, and give more run-time experiments --- see our response below to
> reviewer 1.

---

### Decision · Program_Chairs · 2019-12-19

**Decision:**

Accept (Poster)

**Comment:**

This paper proposes an approach to representing a symbolic knowledge base as a sparse matrix, which enables the use of  differentiable neural modules for inference. This approach scales to large knowledge bases and is demonstrated on several tasks.

Post-discussion and rebuttal, all three reviewers are in agreement that this is an interesting and useful paper. There was intiially some concern about clarity and polish, but these have been resolved upon rebuttal and discussion. Therefore I recommend acceptance.